# The General Population’s Inappropriate Behaviors and Misunderstanding of Antibiotic Use in China: A Systematic Review and Meta-Analysis

**DOI:** 10.3390/antibiotics10050497

**Published:** 2021-04-26

**Authors:** Lixia Duan, Chenxi Liu, Dan Wang

**Affiliations:** School of Medicine and Health Management, Tongji Medical School, Huazhong University of Science and Technology, Wuhan 430030, China; duan_lixia@hust.edu.cn (L.D.); 2019511009@hust.edu.cn (D.W.)

**Keywords:** general population, non-prescription purchase, demand for antibiotics, non-adherence, prophylactic antibiotic use, misunderstanding of antibiotics

## Abstract

The general population has increasingly become the key contributor to irrational antibiotic use in China, which fuels the emergence of antibiotic resistance. This study aimed to estimate the prevalence of the general population’s irrational use behaviors of antibiotics and identify the potential reasons behind them. A systematic review and meta-analysis were performed concerning four main behaviors relevant to easy access and irrational use of antibiotics and common misunderstandings among the population about antibiotics. Four databases were searched, and studies published before 28 February 2021 were retrieved. Medium and high-level quality studies were included. Random effects meta-analysis was performed to calculate the prevalence of the general population’s irrational behaviors and misunderstandings relevant to antibiotic use. A total of 8468 studies were retrieved and 78 met the criteria and were included. The synthesis showed the public can easily obtain unnecessary antibiotics, with an estimated 37% (95% CI: 29–46) of the population demanding antibiotics from physicians and 47% (95% CI: 38–57) purchasing non-prescription antibiotics from pharmacies. This situation is severe in the western area of China. People also commonly inappropriately use antibiotics by not following antibiotic prescriptions (pooled estimate: 48%, 95% CI: 41–55) and preventatively use antibiotics for non-indicated diseases (pooled estimate: 35%, 95% CI: 29–42). Misunderstanding of antibiotic use was also popular among people, including incorrect antibiotic recognition, wrong antibiotic use indication, inappropriate usage, and ignorance of potential adverse outcomes. Over-and inappropriate use of antibiotics is evident in China and a multifaceted antibiotic strategy targeted at the general population is urgently required.

## 1. Introduction

In the past several decades, antibiotic resistance (AR) has been increasingly recognized as one of the biggest threats to public health and economic development worldwide, as it makes common infectious diseases difficult to treat and results in increased mortality and treatment cost [1,2,3]. It has been reported that around 700,000 people die from AR worldwide every year. Without effective countermeasures, this figure is estimated to reach 10 million by 2050, leading to USD 100 trillion economic losses worldwide [4]. In addition to the huge loss to society, patients with AR also require extra hospital stays, additional costs, and have an even greater risk of mortality compared to those without infections [5,6].

To address this issue, a call for whole society engagement from one health perspective is required, covering everyone from all sectors and disciplines who should be responsible for preserving the effectiveness of the antibiotic agents [1]. Among the various stakeholders, the widespread inappropriate and overuse of antibiotics in humans still fuel the emergence of AR [7,8,9]. Though antibiotics are commonly prescription-required medicine for infectious diseases, existing studies have shown that people’s behaviors also play a role contributing to overuse of antibiotics [10,11,12], such as non-prescription purchasing of antibiotics, self-medication of antibiotics, storing and sharing of antibiotics, etc. [13,14,15]. Furthermore, the public’s behaviors can also sway physician’s rational use of antibiotics via expectation and pressure for antibiotics, which has been criticized as one key factor contributing to the unnecessary use of antibiotics by physicians [16,17,18,19,20].

However, people’s irrational use behaviors relevant to antibiotics are prevalent around the world [12,21,22,23]. A systematic review showed that the prevalence of self-medication of antibiotics is around 50% in the southeast Asian region [24]. More than half of pharmacy’s customers purchased antibiotics without a prescription globally and the proportion is much higher in developing countries, such as Indonesia and Ethiopia [19,25]. Furthermore, a recent systematic review showed that around 43% of patients expected antibiotics for respiratory tract infections worldwide [26] and existing studies showed that almost all (96.5%) of these patients are estimated to receive antibiotics [27].

As one of the largest consumers of antibiotics worldwide, China has been long criticized for its over and irrational use. Due to introducing a series of restrictive policies in the past decade, physician’s overuse of antibiotics has been significantly reduced [28]. However, a recent study has shown that in the general population, the demand side of antibiotic misuse increasingly became the key contributors to irrational use of antibiotics [29]. For example, more than half of customers can get antibiotics without prescriptions [30]. More than two thirds of medical students kept a personal stock of antibiotics and nearly one fifth of them used antibiotics as prophylaxis and demanded antibiotics from doctors [10]. It is estimated that roughly 58% of irrational use of antibiotics is due to the general population’s irrational use of antibiotics, compared with physician’s irrational prescribing contributing to the other 42% [29].

To address this issue, it is important to understand to what extent people will behave irrationally when using antibiotics and the potential reasons behind them. In the past decade, increasing studies reported the prevalence of the general population’s irrational use of antibiotics as well as the potential factors contributing to it. Results from these studies showed high degrees of variability of findings, settings, participants, and other study characteristics. Though one recent study has qualitatively synthesized the factors influencing the general population’s irrational use of antibiotics [31], it is still unclear what the prevalence of the general population’s irrational use of antibiotic behaviors is, as well as the prevalence of potential factors contributing to it.

Therefore, this systematic review aimed to synthesize the prevalence of irrational use behaviors as well as potential reasons relevant to antibiotics in China.

## 2. Methods

### 2.1. Review Framework

To understand the general population’s irrational use behaviors and potential reasons behind them, a review framework was developed (Figure 1), following the guidelines from the Joanna Briggs Institute (JBI) regarding the systematic reviews of prevalence and incidence [32]. The protocol of the current study has been registered in PROSPERO (No. CRD42021234432).

Based on the World Health Organization [15], four kinds of behaviors were highlighted which are prevalent among general populations, including demanding antibiotic prescriptions from physicians, non-prescription antibiotic purchasing from pharmacies, non-adherence to antibiotic prescriptions, and prophylactic use of antibiotics. In the current study, these behaviors were classified into two aspects, namely, easy access to antibiotics and irrational use of antibiotics. In the current study, the general population refers to patients and people that are potential patients, for example, residents, the elderly, parents, etc. Prophylactic use of antibiotics refers to the behavior of people taking antibiotics to prevent common infectious diseases, such as the cold or flu, which are mainly non-indicated diseases [33].

In terms of potential reasons for behaviors, the definition of responsible antibiotic use was adopted [34]. To ensure a rational use of antibiotics, people are first required to correctly recognize the common antibiotics, understand the indications antibiotics may help, and, finally, know the appropriate usage and potential adverse outcomes. Thus, the general population’s correct recognition of antibiotics, correct indication of antibiotic use, correct usage of antibiotics, and potential adverse outcomes of antibiotics were treated as four potential reasons for people’s behaviors relevant to antibiotic use.

### 2.2. Search Strategy

Four databases were searched to identify relevant studies published before 28 February 2021, including the China National Knowledge Infrastructure (CNKI), VIP database, WANFANG database, and Chinese Biomedical Literature Service System (CBLSS). The development of search strategies followed participant, intervention, comparison, and outcome (PICOs) principles and search terms included “antibiotic”, “the general population”, “irrational use”, “behavior”, “factor” and their synonyms, combined with Boolean operators (for search examples, see Table 1). Details of search strategies can be found in Appendix A.

### 2.3. Inclusion and Exclusion Criteria

According to the review framework (Figure 1), studies reporting at least one of the four identified general population behaviors regarding antibiotic use were included. We excluded studies concentrating on irrational antibiotic prescribing behavior of healthcare providers. Qualitative studies (no quantitative measures applied), reviews, conference reports, and unavailable full-text research were also excluded. Two reviewers (DL and LC) conducted the screening process independently with a Kappa agreement of 93.1%. Discrepancies were resolved through consensus.

### 2.4. Quality Assessment

Quality of all included studies were assessed on nine criteria based on the “Critical Appraisal Instrument for Studies Reporting Prevalence Data” from the JBI quality evaluation tool (Appendix B) [35]. These criteria concern appropriate sample frame, sample strategy, sample size, description of research settings, coverage of the sample, validity of measurement, data collection, statistical analysis methods, and generalizability of results. Each aspect was assessed by one item, which may be assessed as high quality (scored as 1), low quality, or unclear (scored as 0). The overall quality of each study was calculated, and studies were classified into poor level of quality (0–3), moderate level of quality (4–6), or high level of quality (7–9). Only studies with moderate or high level of quality were further included for meta-analysis. Assessment was conducted by two reviewers (DL and LC) and discrepancies were resolved through consensus.

### 2.5. Data Extraction

A standardized data extraction chart was developed according to JBI’s guidelines (Appendix C) [32]. Three kinds of information were extracted, including basic information of research (author, year of publication, aim of the study), characteristics of methodology (study setting, research type, sampling method, sample size, characteristics of participants), and prevalence of behaviors and potential reasons regarding the general population’s antibiotic use (proportion of the four kinds of irrational antibiotic use behaviors of the general population and proportion of incorrect understanding of the general population relevant to antibiotic use). Different publications that used the same data were extracted once.

### 2.6. Statistical Analysis

The proportion of people who showed irrational use behaviors of antibiotics or who hold misunderstandings towards antibiotic use was synthesized. The prevalence of people’s irrational use behaviors of antibiotics was assessed based on four indicators in the current study, namely, proportion of people who demand antibiotic prescriptions from physicians, proportion of people who purchase antibiotics without prescriptions, proportion of people who do not adherent to antibiotic prescriptions, and proportion of people who prophylactically use antibiotics to prevent the cold or other non-indicated diseases. The prevalence of people’s potential reasons of their irrational use behaviors was assessed based on four aspects, covering the proportion of people who can correctly recognize common antibiotics, the proportion of people who can correctly understand indication for antibiotic use, the proportion of people who can correctly use antibiotics, and the proportion of people who are aware of adverse outcomes of antibiotic use. Data synthesis was conducted using Freeman–Tukey transformed proportions and the pooled estimates were back transformed to ordinary proportions. The random effect meta-analysis with DerSimonian and Laird approach was adopted and the pooled proportions with 95% confidence intervals were calculated.

Subgroup analyses were conducted with respect to the geographical region (eastern, central, western China), study period, type of participant (rural residents, urban residents, students, pharmacy’s customer, parents, and others), quality level (moderate/high), and study size. Q test and I^2^ test were performed to determine the degree of heterogeneity across studies. A multivariable meta-regression analysis was conducted to explore the potential causes of the heterogeneity. The co-variates tested included geographical region, study size, characteristics of participants, study period, and quality of the research. Sensitivity analysis was performed by excluding each study in turn to examine the effect of outliers and confirm the robustness of our findings [36]. All statistical analysis was performed using Stata (version 14.0, Texas: StataCorp).

## 3. Results

### 3.1. Study Selection and Characteristics

A total of 8468 articles were retrieved through the literature search, of which 187 met the inclusion criteria, and a total of 78 studies were included in the final meta-analysis (Figure 2). Almost all (71 out of 78) of the included studies applied cross-sectional design. These studies involved 79,165 participants from eastern (*n* = 39,181), central (*n* = 28,547), and western (*n* = 11,437) areas, covering 25 of the 34 provinces in China. The participants were rural residents (*n* = 11,810), urban residents (*n* = 18,785), rural and urban residents (*n* = 2946), students (*n* = 31,641), children’s parents (*n* = 7153), customers of pharmacies (*n* = 2168), and others (*n* = 4662) (Table 2).

Among the four kinds of irrational use behaviors of people, demanding antibiotic prescription was surveyed by nearly half of the included studies (*n* = 30), followed by non-adherence to antibiotic prescriptions (*n* = 30), non-prescription purchasing (*n* = 29), and prophylactic use of antibiotics (*n* = 20). In terms of potential reasons of these behaviors, roughly half of the included studies surveyed the public’s incorrect recognition of antibiotics (*n* = 39), followed by people’s misconception of incorrect indication (antibiotic killing virus, *n* = 29; antibiotic cures the cold, *n* = 21), incorrect usage (antibiotic courses, *n* = 15; combination antibiotic use, *n* = 21), and potential adverse outcomes (antibiotic resistance, *n* = 36; adverse reaction, *n* = 11).

Studies included were deemed to be at a moderate or high level of quality (score >3). The most common source of bias was from sample coverage (71/78, 91.03%), for which most articles did not report a justification for whether they achieved a sufficient coverage of the identified sample.

### 3.2. The Prevalence of Irrational Use Behaviors of the General Population

#### 3.2.1. Demanding Antibiotic Prescription

The overall pooled estimate indicated over one third (37%, 95% CI: 29–46) of the general population have demanded antibiotic prescriptions from their physicians (Figure 3). Subgroup meta-analysis showed the pooled proportions were significantly different among studies from different regions (*p* < 0.001), different types of participants (*p* < 0.001), and different study sizes (*p* < 0.001). Studies from the western area of China reported the highest pooled proportions (65%, 95% CI: 50–78), followed by the eastern (34%, 95% CI: 26–43), and central areas (26%, 95% CI: 17–37). The highest proportion of behavior for demanding antibiotic prescriptions from physicians was observed in the customers of pharmacies (49%, 95% CI: 47–51), followed by urban residents (48%, 95% CI: 33–64), rural residents (28%, 95% CI: 13–45), students (23%, 95% CI: 13–34), and children’s parents (20%, 95% CI: 12–29) (Table 3).

Substantial heterogeneity was observed among studies (χ^2^ = 7688.51, *p* < 0.001, I^2^ = 99.62%). The multivariable meta–regression yielded a significant multivariate model, explaining more than half of the heterogeneity (*p* = 0.0138, adjusted R^2^ = 55.45%, I^2^ = 93.70%), which indicated that study period (*p* = 0.016) and regions (*p* = 0.001–0.016) significantly influenced the pooled estimate of the general population’s behavior regarding demanding antibiotic prescriptions.

#### 3.2.2. Non–Prescription Antibiotic Purchasing

The overall pooled estimate showed that 47% (95% CI: 38–57) of the general population have purchased antibiotics without prescriptions (Figure 3). A significant difference among the pooled proportions of studies from eastern, central, and western China was observed (*p* < 0.001), with estimates of 34% (95% CI: 25–444), 51% (95% CI: 36–65), and 70% (95% CI: 51–86), respectively. Different types of participants also showed significantly varied pooled proportions of non–prescription antibiotic purchasing behavior (*p* < 0.001), with the highest proportion observed in pharmacy customers (73%, 95% CI: 71–75), followed by urban residents (54%, 95% CI: 32–76), and students (49%, 95% CI: 35–62) (Table 3).

#### 3.2.3. Non–Adherence to Antibiotic Prescription

The general population reported prevalent non–adherence to antibiotic prescriptions (Figure 3), the pooled proportion of this behavior was 48% (95% CI: 41–55). The subgroup meta–analysis showed that no significant difference was identified between studies from eastern (49%, 95% CI: 39–60), central (45%, 95% CI: 35–56), and western (47%, 95% CI: 43–52) China. A significant difference was found between different participants (*p* < 0.001), with the highest pooled proportion of this kind of behavior found in rural residents (67%, 95% CI: 48–84) while the lowest in children’s parents (22%, 95% CI: 18–25). In addition, different qualities of studies also reported different prevalence of non–adherence to antibiotic prescriptions of the public (*p* < 0.001), with high–quality studies reporting a pooled proportion of 63% (95% CI: 61–65) while moderate quality studies reported 50% (95% CI: 43–57) (Table 3).

#### 3.2.4. Prophylactic Use of Antibiotics

The overall pooled estimate demonstrated that 35% (95% CI: 29–42) of the general population have prophylactically used antibiotics to prevent infections (Figure 3). The subgroup meta–analysis showed people from western China reported the highest pooled proportion (47%, 95% CI: 43–51), followed by those from eastern (35%, 95% CI: 25–46), and central China (27%, 95% CI: 18–37). In addition, urban residents reported the highest pooled proportion of prophylactic use of antibiotics, followed by students (34%, 95% CI: 25–44), and rural residents (27%, 95% CI: 16–39) (Table 3).

#### 3.2.5. Sensitivity Analysis

The robustness of these pooled estimates was confirmed by sensitivity analysis. The previously obtained estimates were similar to the results from sensitivity analysis, with non–prescription antibiotic purchasing ranging from 45.86 to 48.64%, demanding antibiotic prescriptions from 36.48 to 39.30%, non–adherence to antibiotic prescriptions from 47.55 to 50.20%, and prophylactic use of antibiotics from 33.19 to 37.53%.

### 3.3. Prevalence of Potential Reasons of the General Population’s Irrational Use of Antibiotics

The proportion of people who showed different kinds of misconceptions of antibiotic use was extracted and their pooled proportions were synthesized, covering antibiotic recognition, correct indication, appropriate usage, and potential adverse outcomes.

The overall pooled proportion showed that nearly 41% (95% CI: 33–49) of the general population were unable to correctly recognize common antibiotics. People were also unable to correctly understand indications for antibiotic use, the pooled estimate of 53% (95% CI: 44–62) of participants believed that antibiotics can kill viruses and 47% (95% CI: 36–58) believed that most colds need antibiotics. People also commonly hold misconceptions about antibiotic usage, a pooled proportion of 44% (95% CI: 31–56) of the population thought that combined use of antibiotics was more effective than use of one antibiotic and 49% (95% CI: 35–64) believed that they should follow antibiotic prescriptions in dosing and duration. In terms of potential adverse outcomes of antibiotic use, the pooled estimates showed that 41% (95% CI: 33–50) of people did not understand that over and inappropriate use of antibiotics leads to antibiotic resistance and 45% (95% CI: 21–71) did not recognize that use of antibiotics may result in adverse reaction (Figure 4).

## 4. Discussion

### 4.1. Main Findings

The infections caused by antibiotic–resistance bacteria are increasingly difficult to treat and the consequences of AR, such as extra hospitalizations and deaths, are self–evident [5,6]. Based on data from 78 studies that included a total of 79,165 participants, this systematic review demonstrates that irrational use behaviors of antibiotics are common among the general population in China. On one hand, antibiotics are easily accessed by the general population. On the other hand, a considerable proportion of people cannot appropriately use antibiotics.

A huge variation of the public’s irrational use behaviors in different geographical regions was also identified based on subgroup analysis. People from western regions of China were more likely to show behaviors resulting in obtaining unnecessary antibiotics. Other than the fact that people reported less demanding of antibiotics from physicians over time, no significant declining trend was identified in people’s irrational use behaviors of antibiotics in the past decades.

The potential reasons for the prevalent over and irrational antibiotic use behaviors of the general population may lie on their misconception of antibiotics, which is also shown to be popular in the current study. It is estimated that roughly half of participants are unable to recognize common antibiotics, understand in what situation antibiotics are required, show sufficient knowledge of appropriate antibiotic usage, and understand adverse outcomes of antibiotics. These misunderstandings may lead to expectations of unnecessary antibiotics from the public and hinder people’s rational use of these medicines.

### 4.2. Strengths and Limitations

In the current study, a review framework was developed based on the WHO’s highlights of the general population’s irrational use behaviors of antibiotics. The definition of rational use of antibiotics was also referred to help identify potential reasons for the general population’s irrational use behaviors. Based on the review framework, a comprehensive search strategy was generated, covering four kinds of common inappropriate use behaviors and four aspects of misunderstanding of antibiotics among the general population in China. In addition, we carefully followed the guidelines of the systematic reviews of prevalence and incidence from JBI. Quality assessment was applied to ensure only studies with high and moderate quality were included. We also performed subgroup analysis and sensitivity analysis to ensure the robustness of the results.

There are also some limitations. First, the quality of the included studies may be underestimated due to the restrictions on the length of text in Chinese journals, causing details of methodology to possibly not be reported [114,115]. Second, only four main datasets were searched and we focused on synthesis of studies published in Chinese. Such strategy was adopted due to the aim of the current study and publication in English has been synthesized in existing reviews. However, the results of the current study seemed not to change even after inclusion of publications in English. Existing reviews synthesized English publications and narratively described prevalence of people’s irrational use behaviors of antibiotics, which is similar when compared with the results in the current study. Finally, there are limited studies on some indicators in the current study, such as the general population’s misconceptions of combination antibiotics use and adverse reaction. More studies are warranted to generate a more accurate and robust estimate.

### 4.3. Comparison with Existing Studies

The prevalence of irrational antibiotic use behaviors among the general population confirmed that they play a role in fueling overuse of antibiotics in China. The results of this study correspond with that from another study synthesizing data from publications in English [31], in which the proportion of people’s non–prescription purchasing behaviors ranged from 8.8 to 84.9% and demanding antibiotic prescriptions ranged from 1.8 to 74.5% in Mainland China. These figures are much higher than those from studies in the USA [20,116,117,118,119] and India [120], in which roughly one fifth of people had asked for antibiotics or obtained antibiotics without prescriptions. Considering only part of the expectation would translate to demanding behaviors, the desire for antibiotics is expected to be higher among the public [16].

It is worth noting that people were less likely to demand antibiotics from physicians over time in China, which was similar when compared with results from a survey in 14 countries that patient antibiotic expectation for respiratory tract infections decreased over time [26]. However, considering this trend was not identified in the public’s non–prescription purchasing behaviors, the decline is likely to be owed to the increasingly restricted regulations on healthcare providers. In past decades, a series of policies and national campaigns were introduced to improve healthcare’s rational use of antibiotics [121,122,123]. The recent national assessment showed that only 10.9% of outpatients end with antibiotic treatment and physicians over–prescribing antibiotics has dramatically decreased in the past 5 years in China [28]. Thus, people may adapt their behaviors and reduce their demanding behavior since it is less likely to result in an antibiotic prescription.

On the other hand, non–prescription purchasing from pharmacies is increasingly become an important outlet to fulfill peoples’ unnecessary demand for antibiotics [29]. This situation seemed stable based on the results from the current study. National simulated client studies showed that over 70% of patients with respiratory tract infections are able to easily obtain unnecessary antibiotics from either offline or online pharmacies in China [124,125]. In addition to high expectations of antibiotics from patients, loose regulation [30], economic incentives [126], fierce market competition, and a fragmentized healthcare system [30] also contributed to this situation. Though several policies have been introduced to enforce the regulation of irregular sales of non–prescription antibiotics after 2014 in China [124], the effect was limited and a comprehensive strategy is warranted [30].

In addition, the association between regional economic development and overuse of antibiotics has been established [127], which implies health inequalities across areas. In the current study, the variation between geographical areas of the public’s behaviors to obtain unnecessary antibiotics corresponds with the results from Scotland, in which the most deprived areas reported the highest antibiotic use. Two main aspects were highlighted contributing to high consumption of antibiotics within such kind of areas, namely, people’s susceptibility to infection [127] (due to poor living conditions, poor diet, etc.) and higher prevalence of misconceptions of antibiotics use [128].

In terms of rational use of antibiotics, the pooled proportion of people who do not follow antibiotic prescription is similar compared with the results in 2006 (44%) [129] and this figure is stable over decades and across regions. The prevalence of people’s non–adherence behavior is higher than results from Italy [130], France [131] and Portugal [132]. Previous studies have shown that adherence was associated with fewer concerns about treatment [133,134]. The traditional perspective to medicines among Chinese population, “as a medicine, it is more or less poisonous”, may contribute to the high proportion of medication non–adherence [135], which is also confirmed in patients with stroke, diabetes mellitus, and rheumatoid arthritis [134]. Prophylactic use of antibiotics is also prevalent among the general population in China, but it is not effective against common viral infections, such as the common cold and flu [29,136]. The results of our study are similar to previous reviews [31]. Studies have shown that preventative antibiotic use was associated with antibiotic storage at home. However, without a comprehensive surveillance system of outlets of antibiotics, antibiotics are commonly stored among families in China [29,137].

The general population’s misconception of antibiotic use is evident in China. In terms of knowledge of indication of antibiotic use, the results in the current study are similar compared with the estimates worldwide that around half of people are not able to correctly answer that antibiotics cannot cure colds and they should not be indicated for viral infection [138]. However, Chinese people seemed less knowledgeable of rational antibiotic usage [135] and are less aware that irrational antibiotic use can result in AR [138]. Though education of the general population is one core measure in China’s national action plan to contain bacterial resistance [139], limited information was available whether this campaign is effective.

### 4.4. Implications

To reduce irrational use of antibiotics, several national campaigns targeted at the public have been implemented worldwide. These interventions focused on educating people on the rational use of antibiotics via printed materials or mass media [140] and were deemed to help reduce antibiotic use in outpatients [140,141]. It has also been shown that multifaceted intervention, both involving physicians and patients, is more effective. To address the prevalent over and irrational use of antibiotics among the public in China, a comprehensive strategy is warranted and several implications may be drawn based on the current study.

First, a more comprehensive surveillance and tracking system of antibiotic prescriptions may be developed. The existing national surveillance system of antibiotic use in China concentrates on public hospitals. Information regarding primary healthcare and other facilities (village clinics, private clinics, etc.) are limited, despite irrational antibiotic being estimated to be more serious in these places. This system also needs to be further linked with pharmacy dispensing data, tracking the end of antibiotic prescriptions, to help irregular sales of antibiotics in both online and offline pharmacies. Through its effect on the general population’s behaviors, this kind of system has been initiated in Jinan, Shandong Province [142]. In addition to surveillance and tracking systems, implementation of a restriction policy of non–prescription sales of antibiotics is also significant. China has announced its national action plan to contain antimicrobial resistance in 2016 and aimed to achieve the goal that antibiotics will only be sold with a prescription in all pharmacies. However, limited supporting measures were identified to help achieve this goal despite non–prescription purchasing still being prevalent [139,143].

In addition, continuous training on antibiotic use and communication skills of physicians is required, especially for those in primary care, private clinics, and prescribing on the internet [30]. Such training programs have been proven to significantly reduce irrational use of antibiotics in children with upper respiratory tract infections in Guangxi Province. Physicians being accountable for prescriptions of antibiotics would reduce the public’s unnecessary access to antibiotics [30] and would further reduce their demand for antibiotics when they experienced similar symptoms the next time [144,145].

Finally, given the effectiveness of several national campaigns worldwide that significantly improve the general population’s awareness of rational antibiotic use [146,147], such interventions may also be introduced in China, particularly in remote regions with poor medical care.

## 5. Conclusions

Irrational use behaviors and misconceptions relevant to antibiotics among the general population is evident in China. People can easily obtain unnecessary antibiotics by demanding them from physicians or purchasing from pharmacies without prescriptions. They also showed popular behaviors of non–adherence to prescription or preventative use for non–indicated diseases, for example, the cold. Roughly half of people are unable to recognize common antibiotics, know the correct indication, understand rational usage, and realize the potential adverse outcomes. To address this issue, a comprehensive strategy is required, including construction of a systematic surveillance and tracking system of antibiotic prescriptions, a training program for physicians of rational use of antibiotics and communication skills, and a national campaign on the general population to improve their awareness.

## Figures and Tables

**Figure 1 antibiotics-10-00497-f001:**
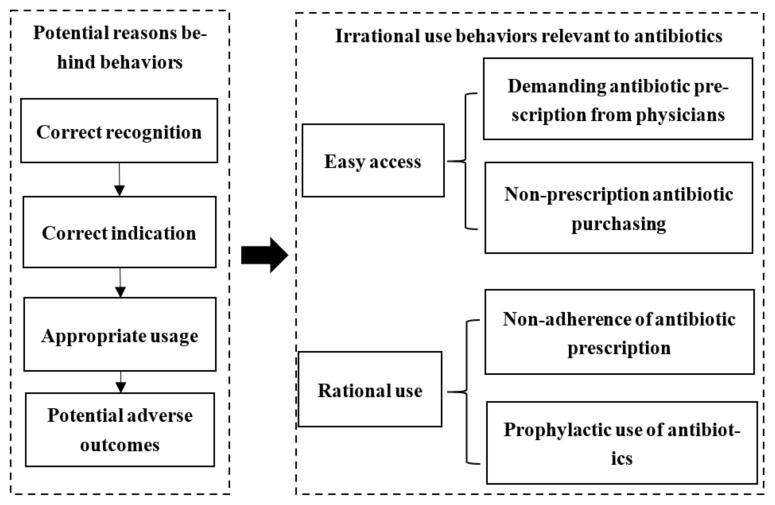
Review framework.

**Figure 2 antibiotics-10-00497-f002:**
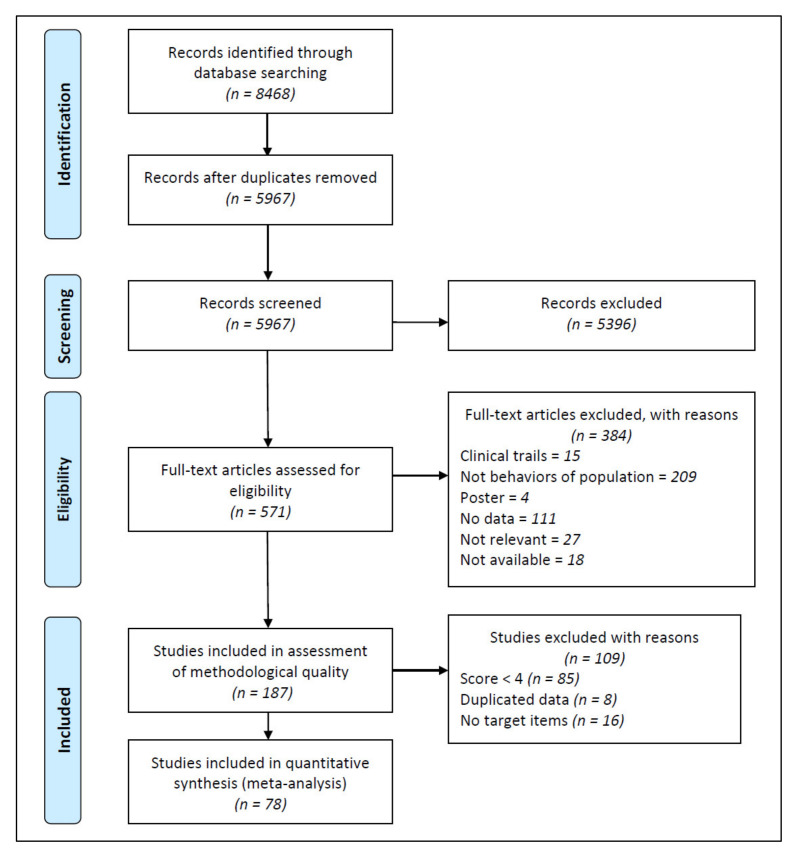
Flow diagram of study identification.

**Figure 3 antibiotics-10-00497-f003:**
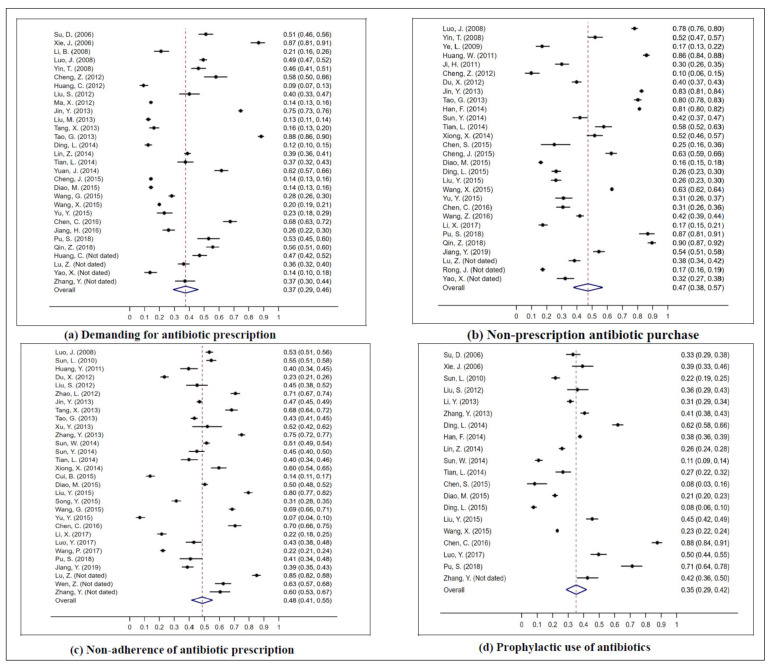
Meta-analysis of the pooled proportion of the general population’s irrational use behaviors regarding antibiotics. (**a**) Demanding for antibiotic prescription. (**b**) Non-prescription antibiotic purchase. (**c**) Non-adherence of antibiotic prescription. (**d**) Prophylactic use of antibiotics.

**Figure 4 antibiotics-10-00497-f004:**
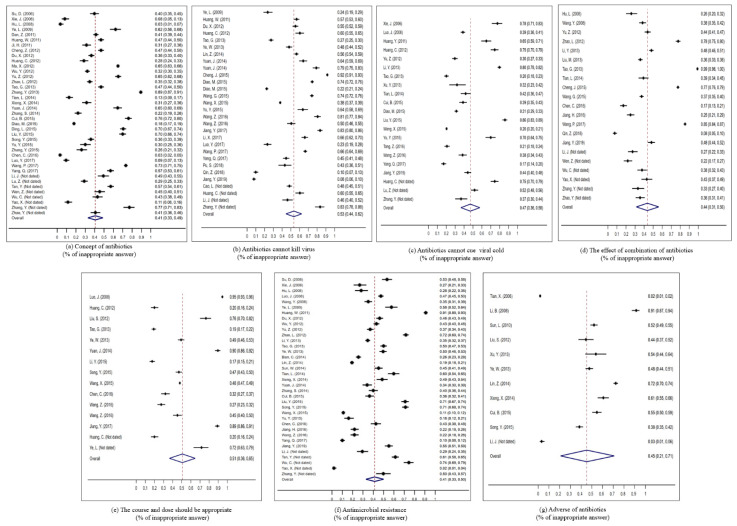
Meta–analysis of the pooled proportion of the general population’s misconceptions regarding antibiotic use. (**a**) Concept of antibiotics (% of inappropriate answer). (**b**) Antibiotics cannot kill virus (% of inappropriate answer). (**c**) Antibiotics cannot cue viral cold (% of inappropriate answer). (**d**) The effect of combination of antibiotics (% of inappropriate answer). (**e**) The course and dose should be appropriate (% of inappropriate answer). (**f**) Antimicrobial resistance (% of inappropriate answer). (**g**) Adverse of antibiotics (% of inappropriate answer).

**Table 1 antibiotics-10-00497-t001:** Search strategy.

#	Searches
1	(antibacterial * or antibiotics * or antimicrobial *).af.
2	(application * or management * or use * or storage * or non-prescri * or self-medication * or irrational * or abuse * or behavior * or status quo *).af.
3	(determinants * or knowledge * or attitude * or belief * or expectation * or medical advice * or compliance *).af.
4	(residents * or population * or general population * or public * or respiratory infection patients * or middle-aged and elderly * or students * or outpatients *).af.
5	1 and 2 and 3 and 4

Note: * is a truncation operator, which refers to words with same stem and different endings to ensure full coverage of study topic.

**Table 2 antibiotics-10-00497-t002:** Characteristics of the included studies.

First Author	Year	Region	Study Design	Type of Participants	Sample Size	Quality Assessment
Tian, X. [37]	2006	East	Cross-sectional	Students	4980	4
Su, D. [38]	2006	East	Quasi-experimental	Rural residents	420	5
Xie, J. [39]	2006	East	Quasi-experimental	Urban residents	209	4
Luo, J. [40]	2008	Central	Cross-sectional	Pharmacy customers	1490	7
Yin, T. [41]	2008	Central	Cross-sectional	Pharmacy customers	396	7
Hu, L. [42]	2008	East	Cross-sectional	Urban residents	209	5
Wang, Y. [43]	2008	East	Cross-sectional	Urban residents	658	4
Li, B. [44]	2008	Central	Cross-sectional	Children’s parents	266	4
Ye, L. [45]	2009	Central	Cross-sectional	Outpatients	276	4
Sun, L. [46]	2010	East	Cross-sectional	Students	853	4
Huang, W. [47]	2011	West	Cross-sectional	Outpatients	956	5
Huang, Y. [48]	2011	East	Cross-sectional	Urban residents	298	4
Ji, H. [49]	2011	West	Cross-sectional	Urban residents	400	5
Shan, H. [50]	2011	East	Cross-sectional	Outpatients	1236	4
Huang, C. [51]	2012	East	Cross-sectional	Rural residents	400	5
Cheng, S. [52]	2012	East	Cross-sectional	Urban residents	164	5
Zhao, L. [53]	2012	East	Cross-sectional	Rural residents	1014	6
Wu, Y. [54]	2012	West	Cross-sectional	Urban and rural residents	1200	5
Ma, X. [55]	2012	East	Cross-sectional	Urban residents	4183	4
Yu, M. [56]	2012	Central	Cross-sectional	Children’s parents	854	5
Liu, S. [57]	2012	East	Cross-sectional	Urban residents	200	4
Du, X. [58]	2012	Central	Cross-sectional	Students	905	4
Tao, G. [59]	2013	West	Quasi-experimental	Urban residents	1000	6
Li, Y. [60]	2013	East	Cross-sectional	Rural residents	1589	5
Jin, Y. [61]	2013	West	Cross-sectional	Urban residents	2556	5
Ye, W. [62]	2013	East	Cross-sectional	Rural residents	800	4
Tang, X. [63]	2013	East	Cross-sectional	Outpatients	496	4
Zhang, Y. [64]	2013	East	Cross-sectional	Students	1180	4
Xu, Y. [65]	2013	East	Cross-sectional	Urban residents	106	5
Liu, M. [66]	2013	East	Cross-sectional	Children’s parents	1046	7
Sun, W. [67]	2014	East	Quasi-experimental	Children’s parents	593	6
Ding, L. [68]	2014	East	Cross-sectional	Children’s parents	722	5
Tian, L. [69]	2014	East	Cross-sectional	Urban residents	316	5
Han, F. [70]	2014	Central	Cross-sectional	Students	4462	5
Yuan, J. [71]	2014	West	Cross-sectional	Rural residents	435	6
Bian, C. [72]	2014	Central	Cross-sectional	Outpatients	1214	4
Zhang, S. [73]	2014	Central	Cross-sectional	Urban residents	500	5
Xiong, X. [74]	2014	East	Cross-sectional	Urban residents	354	4
Lin, W. [75]	2014	East	Cross-sectional	Urban residents	1784	5
Sun, Y. [76]	2014	Central	Cross-sectional	Students	388	4
Liu, Y. [77]	2015	East	Quasi-experimental	Rural residents	639	7
Diao, M. [78]	2015	Central	Cross-sectional	Rural residents	2047	5
Chen, S. [79]	2015	East	Cross-sectional	Teachers	84	8
Wang, X. [80]	2015	Central	Cross-sectional	Students	11,192	5
Li, Y. [60]	2015	Central	Cross-sectional	Students	692	5
Cheng, J. [81]	2015	Central	Cross-sectional	Rural residents	2611	6
Zhang, Y. [82]	2015	East	Cross-sectional	Children’s parents	250	6
Song, Y. [83]	2015	East	Cross-sectional	Students	812	4
Cui, B. [84]	2015	East	Cross-sectional	Students	528	4
Ding, L. [85]	2015	East	Cross-sectional	Rural residents	769	6
Wang, G. [86]	2015	East	Cross-sectional	Urban residents	1718	5
Yu, Y. [87]	2015	East	Cross-sectional	Urban residents	267	5
Lei, Z. [88]	2015	Central	Cross-sectional	Pharmacy customers	282	4
Chen, C. [89]	2016	East	Cross-sectional	Civil servants	400	5
Tang, Z. [90]	2016	West	Cross-sectional	Children’s parents	766	5
Wang, Z. [91]	2016	East	Cross-sectional	Urban and rural residents	438	5
Jiang, H. [92]	2016	East	Cross-sectional	Urban residents	449	6
Wang, Q. [93]	2016	West	Cross-sectional	Students	1584	4
Jiang, Y. [94]	2017	East	Cross-sectional	Urban and rural residents	651	4
Luo, Y. [95]	2017	West	Cross-sectional	Students	368	5
Yang, Q. [96]	2017	East	Cross-sectional	Children’s parents	713	4
Wang, P. [97]	2017	East	Cross-sectional	Students	1957	4
Li, X. [98]	2017	East	Cross-sectional	Children’s parents	658	4
Piao, S. [99]	2018	East	Cross-sectional	Urban residents	181	4
Qin, Z. [100]	2018	West	Cross-sectional	Urban residents	518	4
Jiang, Y. [101]	2019	Central	Cross-sectional	Urban and rural residents	657	6
Cao, L. [102]	NA	East	Cross-sectional	Children’s parents	1080	4
Tan, Y. [103]	NA	West	Cross-sectional	Rural residents	800	4
Ye, L. [104]	NA	East	Cross-sectional	Students	152	6
Yao, X. [105]	NA	East	Cross-sectional	Students	270	6
Zhao, Y. [106]	NA	West	Cross-sectional	Students	363	4
Li, J. [107]	NA	West	Cross-sectional	Rural residents	286	4
Huang, C. [108]	NA	East	Quasi-experimental	Urban residents	400	4
Wu, C. [109]	NA	East	Quasi-experimental	Students	337	4
Wen, W. [110]	NA	Central	Cross-sectional	Urban residents	315	4
Lu, Z. [111]	NA	East	Cross-sectional	Students	600	5
Rong, J. [112]	NA	East	Cross-sectional	Urban residents	2000	4
Zhang, Y. [113]	NA	West	Cross-sectional	Children’s parents	205	4

**Table 3 antibiotics-10-00497-t003:** Subgroup analysis of the proportion of the general population’s irrational use behaviors regarding antibiotics (%, 95% CI).

Subgroup	Demanding Antibiotic Prescription	Non-Prescription Antibiotic Purchasing	Non-Adherence to Antibiotic Prescription	Prophylactic Use of Antibiotics
Geographical region				
Eastern China	34 (26–43)	34 (25–44)	49 (39–60)	35 (25–46)
Central China	26 (17–37)	51 (36–65)	45 (35–56)	27 (18–37)
Western China	65 (50–78)	70 (51–86)	47 (43–52)	47 (43–51)
Study period				
<2010	51 (35–67)	49 (15–83)	53 (51–56)	35 (31–39)
2010–2015	37 (20–56)	57 (43–70)	51 (43–59))	32 (24–40)
≥2015	32 (24–41)	44 (31–58)	40 (27–53)	38 (24–53)
NA	33 (20–47)	29 (15–45)	70 (51–86)	42 (36–50)
Type of participant				
Rural residents	28 (13–45)	32 (15–52)	67 (48–84)	27 (16–39)
Urban residents	48 (33–64)	54 (32–76)	45 (36–55)	39 (26–54)
Students	23 (13–34)	49 (35–62)	43 (27–61)	34 (25–44)
Pharmacy’s customer	49 (47–51)	73 (71–75)	–	–
Children’s parents	20 (12–29)	17 (15–21)	22 (18–25)	–
Urban and rural residents	–	54 (51–58)	39 (35–43)	–
Others	38 (35–41)	40 (7–80)	61 (53–68)	41 (12–74)
Quality score				
4–6	38 (29–47)	48 (37–58)	47 (40–54)	36 (29–43)
7–9	35 (10–64)	45 (18–75)	63 (61–65)	40 (3–44)
Study size				
<400	41 (28–55)	40 (27–53)	44 (34–55)	38 (26–51)
400–800	37 (24–51)	42 (26–58)	54 (31–76)	40 (16–67)
800–1200	88 (86–90)	70 (41–93)	51 (30–72)	32 (30–34)
≥1200	28 (16–42)	54 (34–74)	48 (37–58)	28 (21–34)

## Data Availability

Publicly available datasets were analyzed in this study. This data can be found through the search strategy.

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
