# Peer review of "The General Population’s Inappropriate Behaviors and Misunderstanding of Antibiotic Use in China: A Systematic Review and Meta-Analysis"

_antibiotics, 2021, doi:10.3390/antibiotics10050497_

Round 1
Reviewer 1 Report
This review provides a good overview about an old but ongoing topic with very current reference to the problem of over- and inappropriate use of antibiotics in (different parts of) China.
The research questions are well defined and demarked, the results and outcomes are matching appropriately to these questions.
Analyzing and demarking different reasons as well as variations in proneness of different populations for this topic and it´s far reaching consequences it is very worth to be published.
Its topic addresses the aspects of a generally incorrect understanding of antibiotic target, self-therapy, physician´s prescription and application which is relevant for politicians and health system administrators as well as in health education.
These aspects are very important for fighting AMR from a public health or health care provider perspective and from a patient approach. Educating to reduce antibiotic medication, i.e. in respiratory infections, urinary tract infections or otitis media are goals of many ongoing campaigns.
Other aspects relevant in the bigger picture of AMR could be mentioned (like antibiotics in water circulation or veterinary medicine.
There are only content related necessities of modification.
The authors should explain the problem of antibiotic resistance worldwide and in China more in detail with providing numbers on the phenomenon related to AMR caused illness, hospitalization and death. Explaining the clinical relevance in more detail, not only focussing on the cost problem (Introduction and Discussion).
The clinical impact of AMR has to be illuminated more in detail concerning health risks with some more written statements. From a general public health point of view AMR is the core problem resulting from misapplication of antibiotics endangering whole world societies treatment options for relevant bacterial infections. Besides that there are other important individual adverse effects and risks which also should be mentioned in the paper.
Thus I recommend a description of individual clinical and clinical society risks more detailed in the introduction part and discussion part.
The introduction mentions antimicrobial resistance (AMR) once. For the not quite expert reader, this could be equated with antibiotic resistance. This could be differentiated.
Even if the work looks at a very narrowly defined area, it should be considered whether this section of the problem should be embedded a little more in a larger framework. For example, reference could be made to "the response to AMR through the One Health Global Action Plan", developed by the WHO in collaboration with the Food and Agriculture Organization of the United Nations (FAO) and the World Organisation for Animal Health (OIE). The topic is included as a part of WHO´s initiative within its response plan The 5 objectives of the Global Action Plan on AMR.
Methods and Results are well-presented.
Reviewer 2 Report
Illegal is a quite different concept when compared to inappropriate or irrational use of antibiotics or ‘unnessary’ use. One should be cautious on these ‘words’, that need robust definitions. If I understand the abstract well, antibiotics can be purchased without prescription, so that these words need to be reconsidered.
Along the same line, who is the ‘public’: this reviewer assumes that this refers to the patients, but physicians, pharmacists and manufacturers as well as government are part of this ‘public’ domain. It reads somewhat unbalanced only to focus on ‘the public’ if the environment within this public evolves provides easy access to antibiotics ?
I’m ok on the methodology (systematic search and analysis) as described and applied. It is perhaps useful to add (initials ?) of those who conducted the assessment.
How has ‘prophylactic’ use been defined, and its appropriateness as some (specific, and limited) prophylactic practices are valuable.
The claim that page or word restriction in Chinese journal might explain the limitations on methods should at least be supported by a reference ?
Besides education, there are strong arguments that restrictions related to access to antibiotics (free of prescription) are effective, and this should be further stressed.
Round 2
Reviewer 2 Report
i have verified the rebuttal and the revised version of the paper, and i don't have additional comments or suggestions